# Association between aqueous humor cytokines and postoperative corneal endothelial cell loss after Descemet stripping automated endothelial keratoplasty

**Tatsu Okabe[1], Wataru Kobayashi**  **[1,2]\*, Takehiro Hariya[1], Shunji Yokokura[1], Toru Nakazawa[1,2]**

**1** Department of Ophthalmology, Tohoku University Graduate School of Medicine, Sendai, Japan,
**2** Department of Ophthalmology, Tohoku University Graduate School of Medicine, Retinal Disease Control, Sendai-shi, Miyagi-ken, Japan

\* wkobayashi@oph.med.tohoku.ac.jp

**Data Availability Statement:** All relevant data are within the manuscript and its Supporting information files.

## Abstract

This study measured the intraoperative anterior aqueous humor concentrations of various cytokines during corneal endothelial transplantation and searched for relationships between these concentrations and postoperative corneal endothelial cell (CEC) depletion. We recruited 30 consecutive patients who underwent corneal endothelial transplantation with Descemet's stripping automated endothelial keratoplasty (DSAEK) at Tohoku University Hospital between February 2014 and July 2017. During surgery, we obtained aqueous humor samples and later measured the concentrations of 27 cytokines with a Multiplex Bead Assay (Bio-Plex Pro). We counted CECs 1, 6 and 12 months after surgery, and used Spearman's rank correlation coefficient to identify relationships between CEC depletion and the concentrations of detected cytokines. The loss of CECs 1–6 months after surgery was significantly correlated with IL-7, IP-10, MIP-1a and MIP-1b concentrations (-0.67, -0.48, -0.39, and -0.45, respectively, all P <0.01). CEC loss 1–12 months after surgery was significantly correlated with IL-1b, IL-7, IP-10 and RANTES concentrations (-0.46, -0.52, -0.48, and -0.43, respectively). Multiple regression analysis showed that IL-7 concentration was significantly associated with CEC loss 1–6 months after surgery (b = -0.65, P < 0.01) and IP-10 concentration was associated with CEC loss 1–12 months after surgery (β = -0.38, P < 0.05). These results suggest that not only inflammatory cytokines but also IL-7, a cytokine related to lymphocytes, may be involved in the depletion of CECs after DSAEK, particularly depletion that occurs relatively early.

## Introduction

Descemet's stripping automated endothelial keratoplasty (DSAEK) is now one of the standard treatments for endothelial dysfunction. In comparison with other corneal transplantation methods, such as penetrating keratoplasty (PKP), DSAEK has many advantages [1–4]. DSAEK allows rapid visual recovery [3], fewer graft rejections, and a lower incidence of astigmatism

**Funding:** This work was supported by JSPS KAKENHI Grant Number JP18K09436. The funder had no role in study design, data collection and analysis, decision to publish, or preparation of the manuscript.

**Competing interests:** The authors have declared that no competing interests exist.

after the operation, and also eliminates suture-related complications [5]. Despite these benefits, graft failure after DSAEK is still a major concern. The main cause of graft failure after DSAEK is endothelial decompensation [6, 7]. Endothelial cell density declines with age, with conditions including glaucoma and uveitis, and after intraocular surgery or laser iridotomy for angle-closure glaucoma [8–15]. Other risk factors for the loss of endothelial cells after DSAEK include surgical technique and the environment in the anterior chamber [16]. Several studies have reported that the concentration of cytokines in the anterior aqueous humor is related to the loss of corneal endothelial cells (CECs) after corneal transplantation [17–19]. Other studies have reported that iris damage promotes inflammatory cytokines in the aqueous humor that are related to CEC depletion [20, 21]. However, many points are still unclear. The most common primary disease for corneal transplantation is Fuchs' endothelial corneal dystrophy (FECD) in the U.S., [22, 23] but in Japanese hospitals a great variety of diseases are treated with corneal transplantation. Bullous keratopathy after laser iridotomy is most common [24–26], but pseudophakic bullous keratoconus and FECD are not rare.

Thus, the purpose of this study was to study patients undergoing corneal transplantation at a major Japanese hospital (Tohoku University Hospital), measure the concentration of cytokines in their anterior aqueous humor at the time of corneal endothelial transplantation, and analyze the relationship between cytokine concentration and postoperative CEC depletion.

## Subjects and methods

This retrospective study adhered to the tenets of the Declaration of Helsinki, and the protocols were approved by the clinical research ethics committee of the Tohoku University Graduate School of Medicine. Written informed consent was obtained from all participants included in this study. A total of 30 consecutive patients who underwent DSAEK at Tohoku University Hospital in the period from February 2014 to July 2017 were included in this retrospective longitudinal study. All cases in this study underwent solitary DSAEK. We excluded cases in which CECs could not be measured one month after the operation. Patients with systemic inflammatory diseases such as rheumatoid arthritis, Sjogren's syndrome, and inflammatory bowel disease were excluded. Patients who received systemic steroids were also excluded.

### Surgical technique

All DSAEK surgeries were performed by one of two experienced surgeons (SY, TH). After a circle with a diameter of 8.0 mm was made on the corneal surface, 3 corneal side ports were made for setting up a maintainer and for surgical manipulation in the anterior chamber. Four vertical corneal incisions were made for drainage between the recipient and donor corneas in this circle on the corneal surface. The maintainer was inserted into the anterior chamber and a corneal incision was made with a width of 5.4 mm in the cornea. In patients with FECD, the Descemet membrane was peeled into a circle with the tip of the inverted Sinskey hook. At that time, a small iris incision was created below the iris with a 25-gauge vitreous cutter to prevent air pupillary block. After removing the epithelium of the pre-cut cornea (CorneaGen, Seattle USA), it was trephinated with an 8.0 mm diameter punch (Barron Donor Cornea Punch; Katena Products Inc, Denville, NJ) to create a donor corneal endothelium graft. The donor endothelial lamella was loaded onto a Busin glide (Busin donor glide, Cat #19098; Moria, Antony, France) and then the donor endothelial lamella was pulled through the corneal incision. After donor insertion, nodal suture was performed with 10–0 nylon thread. Air was then injected through the side port and the anterior chamber was completely replaced with air for approximately 15 minutes to allow the graft to adhere to the back of the cornea [27].

## Aqueous humor sampling

Aqueous humor samples were obtained at the beginning of the transplantation surgery under general anesthesia and aseptic conditions. In all cases, the samples (50–100 μL) were collected via paracentesis through the anterior chamber with a 30-gauge needle and a 1.0 mL syringe. The samples were frozen without centrifugation at -80˚C immediately for further analysis. All subjects were prescribed 0.3% gatifloxacin or 1.5% levofloxacin hydrate or 0.5% moxifloxacin hydrochloride hydrate eye drops four times daily for 3 days before surgery. In the patients who underwent cataract surgery before DSAEK, 0.1% betamethasone sodium phosphate and 0.1% bromfenac sodium hydrate eye drops were also prescribed for the surgical eye until the day before the surgery. In patients with FECD or corneal endotheliitis, 0.1% fluorometholone eye drops were prescribed for the surgical eye until the day before the surgery.

## Measurement of cytokine concentration

Frozen aqueous humor samples were analyzed 12 months after DSAEK. We analyzed the concentration of each cytokine with a multiplex bead immunoassay system (Bio-Plex Pro Human Cytokine 27-plex Assay; Bio-Rad Laboratories, Hercules, CA), according to the manufacturer's instructions. We diluted the aqueous samples at a 1:4 ratio, and prepared serial 1:4 dilutions of cytokine standards using the Bio-Plex Human Serum Diluent (Bio-Rad). We calculated the concentration of each cytokine by measuring the intensity of the fluorescence signal in a 50-microliter sample of the diluted aqueous humor. This measurement method has a lower limit of detection of 1 pg/mL per cytokine. The fluorescence intensity of the immunoassay was measured and analyzed with Bio-Plex Manager 6.0 software. The analysis included the following 27 cytokines: IL-1 receptor antagonist (IL-1ra), IL-1b, IL-2, IL-4, IL-5, IL-6, IL-7, IL-8, IL-9, IL-10, IL-12, IL-13, IL-15, IL-17, interferon gamma (IFN-g), fibroblast growth factor (FGF), granulocyte colony-stimulating factor (G-CSF), granulocyte macrophage colony-stimulating factor (GM-CSF), interferon inducible protein-10 (IP-10), monocyte chemotactic protein-1 (MCP-1), macrophage inflammatory protein-1a (MIP-1a), macrophage inflammatory protein-1b (MIP-1b), platelet derived growth factor-β (PDGF-b), macrophage inflammatory protein-1 (MIP-1), CC chemokine ligand-3 (Eotaxin), regulated on activation, normal T-cell expressed and secreted (RANTES), T-cell necrotizing factor-a (TNF-a), and vascular endothelial growth factor (VEGF). The assay had a different reliable working range for each cytokine (this information was provided by the manufacturer) [28].

## Data analysis

Ophthalmologic data included spherical equivalent, intraocular pressure, and best-corrected distance visual acuity (BCVA). Decimal BCVA was converted to logMAR BCVA. A specular microscope examination and a slit-light evaluation were performed preoperatively and at 1, 6, and 12 months postoperatively. We counted the number of CECs with a specular microscope (SP-3000P; Topcon, Tokyo, Japan) and analyzed the average cell density by counting about 50 cells. We also recorded donor information including donor age, time between death and preservation, and time between preservation and surgery, and analyzed these data statistically.

## Statistical analysis

The Anderson-Darling test was used to measure whether the data fit a normal distribution. The Kruskal Wallis test was used to compare the CEC numbers and the Spearman's correlation analysis was used to determine the correlation between endothelial cell counts 1, 6, and 12 months after surgery and cytokine concentrations. The Welch's test and the Wilcoxon signed-

rank test were used to determine differences between groups of patients with and without iris damage. A multiple regression analysis was used to determine the correlation between the loss of CECs and cytokines. Our statistical analysis relied on the JMP Pro version 9.0.2 software for Windows (SAS Institute, Japan). P values < 0.05 were considered to be statistically significant.

## Results

Patient data are shown in Table 1. There were significant differences between preoperative visual acuity and visual acuity at 1, 6, and 12 months postoperatively (all P < 0.0001). The etiologies of DSAEK in the studied eyes included pseudophakic bullous keratopathy, post-laser iridotomy bullous keratopathy, FECD, post-trabeculectomy bullous keratopathy, chronic angle-closure glaucoma, corneal endotheliitis, and trauma (Table 2).

The donor corneas were all imported. The mean donor age was 60.5 ± 8.9 years, the mean time between death and preservation was 762.3 ± 364.4 minutes, and the mean time between preservation and surgery was 6.5 ± 0.5 days. The mean number of donor CECs was 2747.5 ± 217.0 cells/mm$^2$. The average number of CECs 1, 6, and 12 months after DSAEK is shown in Table 3 and Fig 1, with the average decreases in CEC counts from months 1 to 6, months 1 to 12, and months 6 to 12. There was no significant correlation between CEC loss at 1, 6, and 12 months postoperatively and donor age (P = 0.63, 0.21, 0.42, respectively), time between death and preservation (P = 0.42, 0.51, 0.35, respectively), time between preservation and surgery (P = 0.26, 0.64, 0.59, respectively), or mean number of donor CECs (P = 0.09, 0.10, 0.10, respectively). There were 17 eyes with iris damage and 13 eyes without iris damage. We compared the number of CECs after surgery in groups of patients with and without iris damage at postoperative months 1, 6, and 12, and the changes in CEC count from months 1 to 6, 1 to 12, and 6 to 12. We found that there were no significant differences between the two groups (Table 4, Figs 2 and 3). Table 5 shows cytokine concentrations of the aqueous humor.

**Table 1. Study characteristics.**

| | |
|---|---|
| Age (years) | 78.6 ± 9.61 |
| Sex ratio | M/F = 11/19 |
| IOL (eyes) | 30 |
| logMAR BCVA preoperatively | 0.89 ± 0.43 |
| logMAR BCVA at 1M | 0.44 ± 0.26 |
| logMAR BCVA at 6M | 0.29 ± 0.22 |
| logMAR BCVA at 12M | 0.28 ± 0.27 |

IOL: intraocular lens, BCVA: best-corrected distance visual acuity.

**Table 2. Etiologies of bullous keratopathy.**

| Primary disease | Cases |
|---|---|
| Laser iridotomy | 12 |
| Fuchs' endothelial corneal dystrophy | 6 |
| Cataract surgery/trabeculectomy | 4 |
| Chronic angle-closure glaucoma | 4 |
| Corneal endotheliitis | 3 |
| Trauma | 1 |
| Total | 30 |

**Table 3. CEC count 1, 6 and 12 months after DSAEK and inter-period loss.**

| Time point | Mean CEC count ± SD |
|---|---|
| Month 1 | 1927.7 ± 429.2 |
| Month 6 | 1611.8 ± 709.3 |
| Month 12 | 1365.3 ± 750.6 |
| Month 1 to month 6 | -315.9 ± 512.6 |
| Month 1 to month 12 | -562.5 ± 623.7 |
| Month 6 to month 12 | -246.5 ± 429.9 |

CEC: corneal endothelial cell; SD: standard deviation

We also compared the aqueous humor concentration of 12 cytokines in patients with and without iris damage: IL-1b, IL-1ra, IL-6, IL-7, IL-8, IL-13, Eotaxin, IP-10, MCP-1, MIP-1a, MIP-1b, and RANTES; there were no significant differences between the two groups (Table 6). On the other hand, we found that the preoperative concentrations of IL-7, IP-10, MIP-1a, and MIP-1b in the aqueous humor were associated with postoperative CEC depletion, and that these four cytokines showed a negative correlation with the change in CEC count from months 1 to 6 (Fig 4).

We also found that IL-1b, IL-7, IP-10, and RANTES were correlated with CEC loss and showed a negative correlation with the change in CEC count from months 1 to 12 (Fig 5). Furthermore, the discrimination analysis showed that IL-7 was the strongest contributing factor to CEC depletion from months 1 to 6, while IP-10 was the strongest contributing factor from months 1 to 12 (Tables 7 and 8).

## Discussion

This study examined multivariate correlations between preoperative cytokine concentrations in the anterior aqueous humor and CEC depletion after corneal transplantation at our institution. We found significant correlations between the rate of CEC loss and IL-7, IP-10, MIP-1a, and MIP-1b at six months after surgery and also IL-1b, IL-7, IP-10, and RANTES one year after surgery. We also found that patients with and without iris damage did not significantly

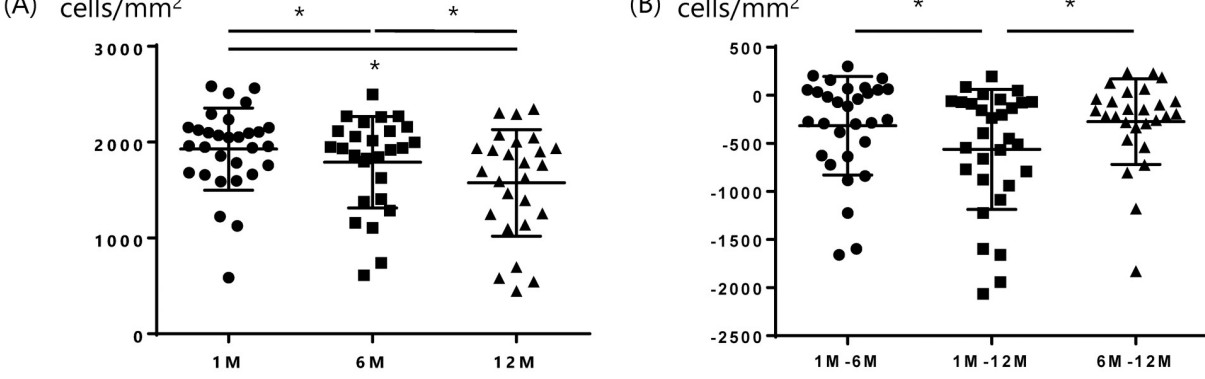

**Fig 1. Comparison of cornea endothelial cell (CEC) numbers and depletion of CEC numbers in groups at months 1, 6, and 12.** (A) and (B) show CEC numbers and depletion of CEC numbers. Kruskal-Wallis Test was used to examine the differences and p-values less than 0.05 were regarded as statistically significant. There were significant differences between them. CEC: Corneal endothelial cell, Kruskal-Wallis test, *P < 0.05.

**Table 4. CEC counts after surgery in groups with and without iris damage.**

| Time point | Iris damage (-): 13 eyes | Iris damage (+): 17 eyes | P value |
|---|---|---|---|
| | CEC count (mean ± SD) | CEC count (mean ± SD) | |
| 1M† | 2088.7 ± 280.8 | 1832.8 ± 390.4 | 0.07 |
| 6M†† | 1860.3 ± 424.4 | 1327.8 ± 867.0 | 0.07 |
| 12M†† | 1609.6 ± 561.9 | 1086.1 ± 857.5 | 0.11 |
| 1M to 6M†† | -228.4 ± 365.5 | -416 ± 641.6 | 0.45 |
| 1M to 12M†† | -479.1 ± 564.7 | -657.7 ± 693.8 | 0.68 |
| 6M to 12M† | -250.8 ± 406.4 | -241.7 ± 470.7 | 0.87 |

CEC: Corneal endothelial cell, SD: Standard deviation,

† Welch's t test,

†† Wilcoxon signed-rank test

differ in the rate of postoperative CEC decrease or in cytokine concentrations in the anterior aqueous humor.

Changes in cytokine concentrations in the anterior aqueous humor have been reported to be associated with the etiology of various ocular diseases and intraocular conditions, including

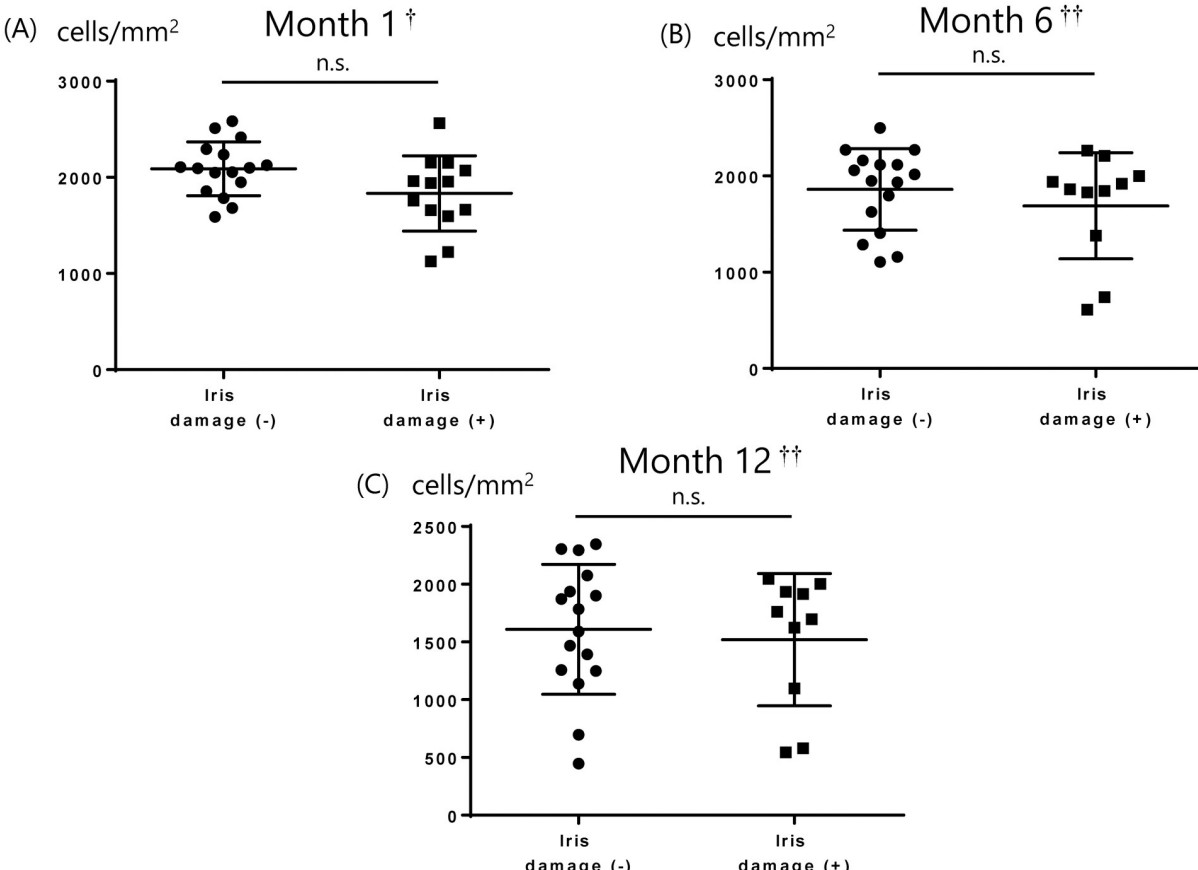

**Fig 2. Comparison of CEC numbers in groups with and without iris damage at months 1, 6, and 12.** Welch's test and Wilcoxon signed-rank test were used to examine the differences and p-values less than 0.05 were regarded as statistically significant. There were no significant differences between them. CEC: Corneal endothelial cell, n.s.: Not significant, † Welch's t test, ††Wilcoxon signed-rank test.

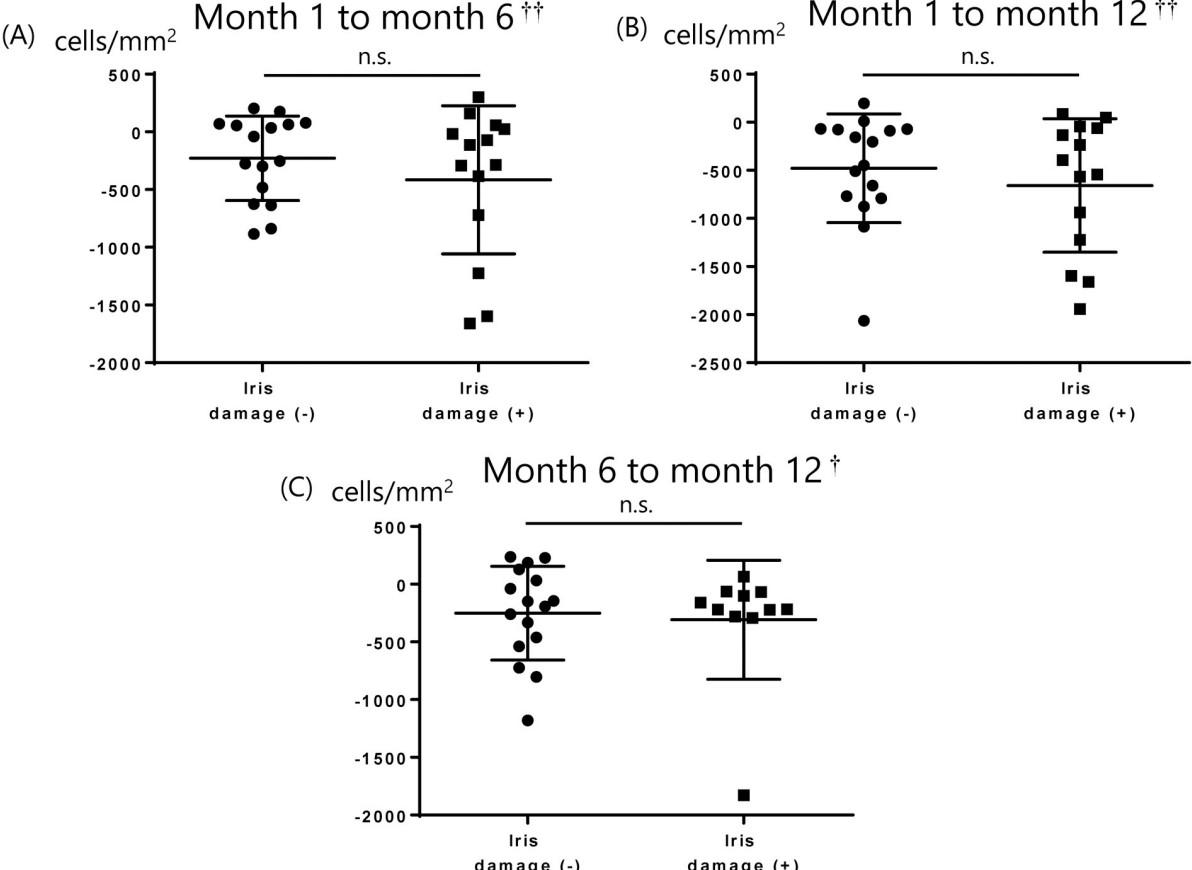

**Fig 3. Comparison of CEC depletion in groups with and without iris damage from months 1 to 6, 1 to 12, and 6 to 12.** Welch's test and Wilcoxon signed-rank test used to examine the differences and p-values less than 0.05 were regarded as statistically significant. There were no significant differences between them. CEC: Corneal endothelial cell, n.s.: Not significant, †Welch's t test, ††Wilcoxon signed-rank test.

FECD, graft rejection, uveitis, glaucoma, AMD, and others [19]. Intraocular surgeries, such as for cataract, and ocular inflammatory diseases, such as uveitis, are also risk factors for CEC loss, but the exact mechanism remains unknown. It is likely that the decrease in CECs is caused by both the primary disease and alterations in the environment of the anterior aqueous humor caused by invasive surgery. It has been reported that remodeling of anatomical structures in the eye continues for five years after DSAEK [29], and post-operative effects are therefore likely to be long-lasting.

IL-7 is produced by the stromal cells of many tissues and is involved in processes such as cell survival, proliferation, and differentiation [30, 31], and also contributes to T-cell proliferation and survival [32]. It has been reported that CECs inhibit the proliferation of CD4-positive and CD8-positive T-cells, and that galectin 9, which is present in endothelial cells, promotes the conversion of CD8-positive T-cells to regulatory T-cells [33, 34]. It has also been reported that IL-7 signaling induces the expression of the co-receptor CD8 and promotes the binding of T-cell receptors and MHC-I molecules [35]. These reports suggest that the reduction of CECs, due to various diseases, induces a relative decrease in galectin 9, increase in CD8-positive T-cells, and a decrease in regulatory T-cells. We consider that increased IL-7 is also likely involved in this cascade, resulting in post-transplantation CEC depletion caused by CD8-positive T-cells.

**Table 5. Cytokine concentrations in the aqueous humor.**

| Cytokine | pg/ml (mean ± SD) | Cytokine | pg/ml (mean ± SD) |
|---|---|---|---|
| IL-1b | 0.07 ± 0.16 | Eotaxin | 12.9 ± 22.2 |
| IL-1ra | 686.9 ± 792.4 | FGF | U.D. / N.D. |
| IL-2 | U.D. / N.D. | G-CSF | U.D. / N.D. |
| IL-4 | 0.58 ± 0.98 | GM-CSF | U.D. / N.D. |
| IL-5 | U.D. / N.D. | IFNg | U.D. / N.D. |
| IL-6 | 713.4 ± 2048.5 | IP-10 | 3172.8 ± 5313.9 |
| IL-7 | 9.72 ± 14.5 | MCP-1 | 760.6 ± 476.1 |
| IL-8 | 149.6 ± 133.5 | MIP-1a | 1.18 ± 1.40 |
| IL-9 | U.D. / N.D. | PDGF-b | U.D. / N.D. |
| IL-10 | U.D. / N.D. | MIP-1b | 10.6 ± 24.3 |
| IL-12 | U.D. / N.D. | RANTES | 38.3 ± 73.8 |
| IL-13 | U.D. / N.D. | TNF-a | U.D. / N.D. |
| IL-15 | U.D. / N.D. | VEGF | U.D. / N.D. |
| IL-17 | U.D. / N.D. | | |

IL- 1b, 2, 4, 5, 6, 7, 8, 9, 12, 13, 15, 17: interleukin-1-beta, 2, 4, 5, 6, 7, 8, 9, 12, 13, 15, 17, IL- 1ra: interleukin-1 receptor antagonist, FGF: fibroblast growth factor, G-CSF: granulocyte colony-stimulating factor, GM-CSF: granulocyte-macrophage colony-stimulating factor, IFN-g: interferon gamma, IP-10: interferon-inducible protein 10, MCP-1: monocyte chemotactic protein-1, MIP-1a: Macrophage inflammatory protein 1-alpha, PDGF-b: platelet-derived growth factor-beta, MIP-1b: macrophage inflammatory protein 1-beta, RANTES: regulated on activation, normal T-cell expressed and secreted, TNF-a: tumor necrosis factor-alpha, VEGF: vascular endothelial growth factor

MIP-1a and MIP-1b are produced and secreted by various cells, especially macrophages, dendritic cells, and lymphocytes. Their production can be induced by many proinflammatory cytokines, such as TNF-a, IFN-g, and IL-1b [36]. They belong to the CC family of chemokines, which show chemotactic activity against leukocytes such as neutrophils, monocytes and lymphocytes, and play an important role in the inflammatory response [37]. MCP-1 and RANTES also belong to the CC family [38]. Inflammation and immune responses of the body to various

**Table 6. Comparison of cytokine concentrations in the aqueous humor in patients with and without iris damage.**

| Cytokine | Iris damage (-): 13 eyes | Iris damage (+): 17 eyes | P value |
|---|---|---|---|
| | pg/ml (mean ± SD) | pg/ml (mean ± SD) | |
| IL-1b | 0.10 ± 0.22 | 0.03 ± 0.04 | 0.49 |
| IL-1ra | 696.1 ± 836.2 | 676.3 ± 770.6 | 0.74 |
| IL-6 | 700.4 ± 1880.2 | 728.4 ± 2298.1 | 0.53 |
| IL-7 | 2.8 ± 4.6 | 17.6 ± 18.6 | 0.06 |
| IL-8 | 133.3 ± 150.2 | 168.3 ± 114.1 | 0.19 |
| IL-13 | 0.78 ± 1.67 | 0.45 ± 0.25 | 0.37 |
| Eotaxin | 10.0 ± 23.9 | 16.4 ± 20.3 | 0.09 |
| IP-10 | 1651.5 ± 1244.9 | 4911.4 ± 7422.9 | 0.43 |
| MCP-1 | 607.3 ± 375.1 | 935.8 ± 530.5 | 0.07 |
| MIP-1a | 0.84 ± 0.67 | 1.59 ± 1.89 | 0.36 |
| MIP-1b | 4.9 ± 4.8 | 17.1 ± 34.7 | 0.23 |
| RANTES | 42.1 ± 92.3 | 33.9 ± 47.8 | 0.93 |

IL- 1b, 6, 7, 8, 13: Interleukin-1 beta, 6, 7, 8, 13, IL- 1ra: Interleukin-1 receptor antagonist, IP-10: Interferon-inducible protein 10, MCP-1: Monocyte chemotactic protein-1, MIP-1a: Macrophage inflammatory protein 1-alpha, MIP-1b: Macrophage inflammatory protein 1-beta, RANTES: Regulated on activation normal T-cell expressed and secreted, SD: Standard deviation, Wilcoxon signed-rank test was conducted

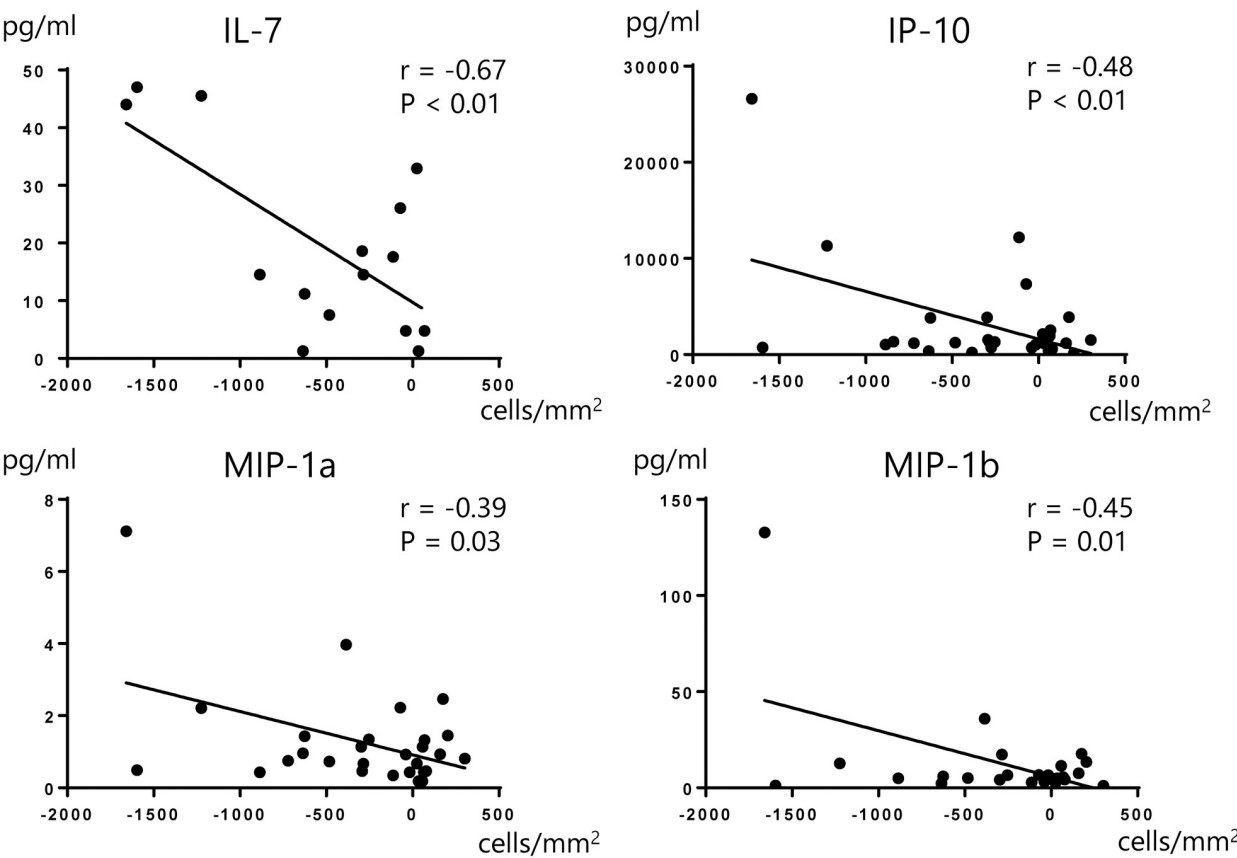

**Fig 4. Correlation of cytokine density and CEC depletion from months 1 to 6.** A correlation analysis was conducted by calculating the Spearman's correlation coefficient and p-values less than 0.05 were regarded as statistically significant. IL-7, IP-10, MIP-1a and MIP-1b were significantly correlated to CEC depletion from months 1 to 6. CEC: Corneal endothelial cell, IL-7: Interleukin 7, IP-10: Interferon-inducible protein 10, MIP-1a: Macrophage inflammatory protein 1-alpha, MIP-1b: Macrophage inflammatory protein 1-beta.

stresses cause infiltration of specific leukocytes into local organs; this infiltration of immune cells leads to the inflammatory response. MIP-1a and MIP-1b regulate the activation and migration of leukocytes to tissues and are known to increase in acute inflammation [37, 38]. IP-10 is secreted from monocytes, endothelial cells and fibroblasts in response to IFN-g [39]. Several roles are attributed to it, including the chemoattraction of monocytes, macrophages, T-cells, NK cells and dendritic cells, the facilitation of T-cell adhesion to endothelial cells, anti-tumor activity, bone marrow colony formation, and the inhibition of angiogenesis [40, 41]. IP-10 acts on a variety of cells, and especially on endothelial cells. It has been reported that IP-10 has an antiproliferative effect in vitro [42] and that it plays an important role in Th1 cell-derived inflammatory and autoimmune diseases, such as Hashimoto's disease, Graves' disease, and type 1 diabetes mellitus [43, 44]. Our results suggest that not only IL-7 but also these acute inflammation-related cytokines contribute to reduced CEC numbers at six months after corneal transplantation. These results are consistent with previous reports.

IL-1b is hardly detectable in normal tissues; it is produced and secreted by immune system cells, such as macrophages, that are activated by the inflammatory response [45]. Gene expression of IL-1b is induced by transcription factors such as NF-kB, which is activated by inflammation-inducing stimulation, and matures with RNA processing by caspase-1 [45, 46]. IL-1b also acts on neutrophils to promote superoxide production, migration and degranulation (as a

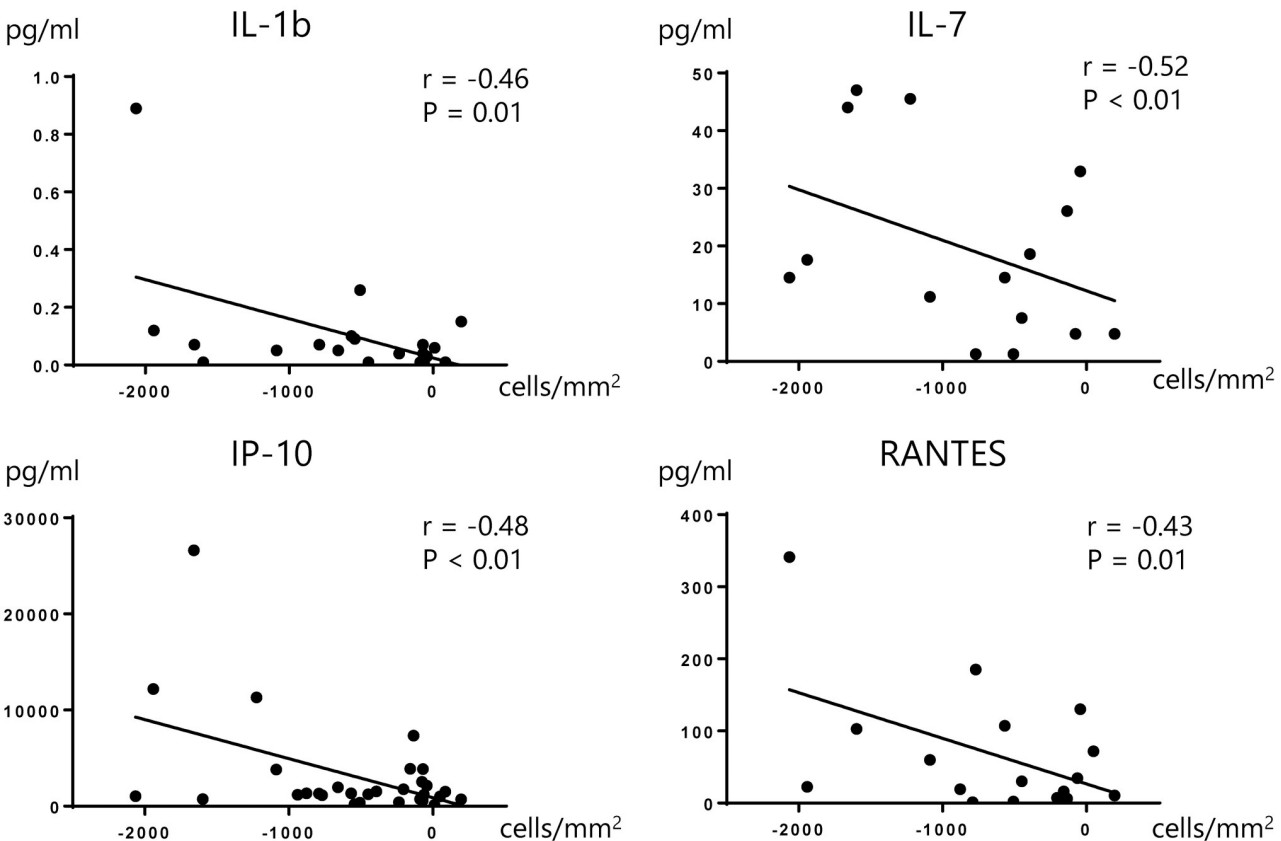

**Fig 5. Correlation of cytokine density and CEC depletion from months 1 to 12.** A correlation analysis was conducted by calculating the Spearman's correlation coefficient and p-values less than 0.05 were regarded as statistically significant. IL-1b, IL-7, IP-10 and RANTES were significantly correlated to CEC depletion from months 1 to 12. CEC: Corneal endothelial cell, IL-7: lnterleukin-7, IP-10: Interferon-inducible protein 10, RANTES: Regulated on activation normal T-cell expressed and secreted.

priming effect), and directly stimulates superoxide production (a triggering effect) [47]. In addition, it has been reported that IL-1b inhibits the function of regulatory T-cells and may thereby contribute to the pathogenesis of tissue damage [48]. RANTES is involved in the chemotaxis of T-cells, eosinophils, and basophils, and plays an active role in mobilizing leukocytes to areas of inflammation [49]. It is secreted from activated CD8-positive T-cells and other immune cells along with the related chemokines MIP-1a and MIP-1b. In addition, it has

**Table 7. Discrimination analysis of factors independently contributing to CEC depletion.** CEC loss from months 1 to 6.

| Variable | B | Std. Error | Beta | T value | P value | VIF |
|---|---|---|---|---|---|---|
| Intercept | -123.9 | 116.2 | 0 | -1.07 | 0.29 | |
| IL-7 | -22.4 | 6.2 | -0.65 | -3.61 | 0.001 | 1.65 |
| MIP-1b | -11 | 8.2 | -0.52 | -1.34 | 0.19 | 7.62 |
| MIP-1a | 70.3 | 126.2 | 0.19 | 0.56 | 0.58 | 6.01 |
| IP-10 | 0.02 | 0.03 | 0.19 | 0.71 | 0.48 | 3.75 |

CEC: Corneal endothelial cell, IL-7: Interleukin 7, IP-10: Interferon-inducible protein 10, MIP-1a: Macrophage inflammatory protein 1-alpha, MIP-1b: Macrophage inflammatory protein 1-beta, VIF: Variance inflation factor

**Table 8. Discrimination analysis of factors independently contributing to CEC depletion.** CEC loss from months 1 to 12.

| Variable | B | Std. Error | Beta | T value | P value | VIF |
|---|---|---|---|---|---|---|
| Intercept | -165.3 | 110.3 | 0 | -1.5 | 0.15 | |
| IP-10 | -0.05 | 0.02 | -0.38 | -2.06 | 0.04 | 1.93 |
| IL-1b | -1221.9 | 735.7 | -0.32 | -1.66 | 0.11 | 2.1 |
| IL-7 | -9.89 | 7.91 | -0.24 | -1.25 | 0.22 | 1.99 |
| RANTES | -1.85 | 1.78 | -0.21 | -1.04 | 0.31 | 2.48 |

CEC: corneal endothelial cell; IP-10: interferon-inducible protein 10; IL-1b: interleukin-1 beta; RANTES: regulated on activation, normal T-cell expressed and secreted; VIF: variance inflation factor

strong migratory activity against immune cells and is expressed over a long period, especially during the acute and chronic phases of disease [50, 51]. Our results suggest that not only IL-7, but also IL-1b, IP-10, and RANTES, which are representative inflammatory cytokines, were significantly correlated with the reduction in CEC numbers one year after surgery. These results show that the combination of IL-7, inflammatory cytokines and CD8-positive T-cells is likely to be related to the decrease in CECs. Multiple regression analysis suggested that IL-7-related CEC depletion was significantly involved in the first six months after surgery, while the inflammatory cytokine IP-10 had a greater effect over the longer term.

Our study had several limitations. Previous reports have shown that iris damage is associated with CEC loss after CEC transplantation surgery [7, 21]. We were unable to replicate this result, possibly because the patients with iris damage in this study had only slight damage in the first quadrant, caused by laser iridotomy, which had been performed many years before. The iris-damage group included 16 eyes with damage in one quadrant (laser iridotomy: 11 eyes, peripheral iridectomy: 4 eyes, iris atrophy: 1 eye) and 1 eye with damage in two quadrants (laser iridotomy and peripheral iridectomy). Therefore, it is unlikely that iris damage affected CEC loss. In future studies, we would like to include cases with more extensive iris damage, such as that caused by mature cataract, post-iris adhesion, or intraoperative floppy iris syndrome. Moreover, inflammatory cytokines, which have been previously reported to be elevated, did not show a significant association in this study. This may be due to differences in the breakdown of the underlying diseases, surgical technique, and the measurement kit. It is also possible that although the samples were stored at -80°C after sampling, they deteriorated due to the time delay before measurements were taken. More timely testing of the samples and the use of a variety of measurement kits may allow for more accurate investigation in the future. In addition, we consider that a future study must evaluate T-cell proliferation using flow cytometry to investigate the increase in CD8-positive T-cells. It is known that the expression of TGFb, which is necessary for the regulation of CD8-positive T-cells, is reversely regulated by IL-7 [52]. Therefore, the verification by measurement of TGFb concentration is required in the future. It is known that the aqueous humor of eyes with iris atrophy contains neutrophils and complements [53]. This study used aqueous humor samples that did not undergo centrifugation, which may have affected our results. It is necessary to compare this study with a follow-up study that analyzes the centrifugation-derived supernatant. Although this study had a number of limitations, our findings suggest that in addition to inflammatory cytokines, IL-7 is also involved in the decrease in CECs after corneal endothelium transplantation. This suggests that CEC decrease is caused not only by acute inflammation, which primarily involves macrophages and neutrophils, but also chronic inflammation, to which lymphocytes make the primary contribution.

## Conclusions

In this study, we suggest it is possible that not only inflammatory cytokines but also IL-7, a lymphocyte-associated cytokine, may be responsible for the post-DSAEK depletion of CECs, and specifically the depletion which takes place at a relatively early stage.

## Supporting information

**S1 Data.**
(XLSX)

## Acknowledgments

The authors thank Mr. Tim Hilts for reviewing this manuscript and thank all the technical staff at the Department of Ophthalmology of Tohoku University for their assistance and encouragement.

## Author Contributions

**Investigation:** Shunji Yokokura.

**Project administration:** Takehiro Hariya, Shunji Yokokura.

**Supervision:** Toru Nakazawa.

**Writing – original draft:** Tatsu Okabe.

**Writing – review & editing:** Wataru Kobayashi.

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
