## [Decision Letter · Decision Letter 0]

14 Oct 2021

PONE-D-21-21341Association between anterior aqueous humor cytokines and postoperative corneal endothelial cell depletion after corneal endothelial cell transplantationPLOS ONE

Dear Dr. Kobayashi,

Thank you for submitting your manuscript to PLOS ONE. After careful consideration, we feel that it has merit but does not fully meet PLOS ONE’s publication criteria as it currently stands. Therefore, we invite you to submit a revised version of the manuscript that addresses the points raised during the review process. 

Specifically, address the missing information and citations along with correcting grammar errors. 

We look forward to receiving your revised manuscript.

Kind regards,

Andrew W Taylor, Ph.D.

Academic Editor

PLOS ONE

Journal Requirements:

a) Did participants provide their written or verbal informed consent to participate in this study?

This study was supported in part by research grants from Japan Science and Technology Agency Center for Revitalization Promotion. 

Reviewers' comments:

Reviewer's Responses to Questions

**Comments to the Author**

1. Is the manuscript technically sound, and do the data support the conclusions?

Reviewer #1: Yes

2. Has the statistical analysis been performed appropriately and rigorously? 

Reviewer #1: Yes

3. Have the authors made all data underlying the findings in their manuscript fully available?

Reviewer #1: Yes

4. Is the manuscript presented in an intelligible fashion and written in standard English?

Reviewer #1: Yes

5. Review Comments to the Author

Reviewer #1: Okabe T et al. evaluated aqueous cytokine levels in the anterior chamber and their correlation with corneal endothelial cells (CEnCs) after DSAEK. This is a well-conducted interesting study and will bring us important information. I think the current manuscript is well-written, however, please consider revising it following the comments below.

Minor comments

1. Title: Please consider revising the title. Title should be simple and specific. The current one is confusing. For example, "corneal endothelial cell transplantation" can be interpreted as "DEMK", "DSAEK", "CEnC cell injection therapy" or "DLEK"... In this case, the authors performed DSAEK, so please specify "Descemet stripping automated endothelial keratoplasty". "anterior aqueous humor" is also confusing. In this article, it is "aqueous humor", as in the text. "reduction" or "loss " would be better than "depletion".

2. Introduction. Well-written with extensive literature review. The hypothesis and purpose are clearly described.

3. Methods: Please clarify whether all patients underwent solitary DSAEK or some of them underwent DSAEK combined with cataract surgery. If so how many?

4. Tables 1 and 2 can be deleted. The data is well described in the text. The title of table 2 would be "etiologies of bullous keratopathy". "Breakdown" sound strange.

5. Methods Line 119. Aqueous sample was frozen without spinning down cellular component, or was it supernatant? Please clarify. Because it sometimes contain immune cells of patients with bullous keratopathy, such as neutrophils, as reported in the article below.

Pathological processes in aqueous humor due to iris atrophy predispose to early corneal graft failure in humans and mice. Sci Adv. 2020 May 13;6(20):eaaz5195.

6. Methods: Please add "Data analysis" paragraph, which clarify how the authors defined "iris damage", how they measured "ECD", how they measured VA and so on. "iris damage" is one of the main outcome. In the past reports, iris damage needs careful evaluation. It can be divided into "iris depigmentation", "iris defect" "iris atrophy" and "laser iridotomy". Some previous reports showed mild iris damage (IDS1-2) did not have any influence on CECD after DSAEK, whereas severe iris had serious effect on it. Thus, as in Table 4 and Figures, comparing subjects stratifying them based on the presence of iris damage is not proper, I think. However, the results in the current study is true. So, please add one paragraph in the Discussion section on this matter.

7. Methods: Please add the information on the donor cornea, domestic or imported, donor age, time between death to preservation, and preservation to surgery if it is available.

8. Results "table 3 title". Please consider rephrasing "inter-period LOSS" not "depletion".

9. Discussion is well written. Line 334 "previous reports" need references. The authors found the significant correlations among CECD and cytokines such as IL-7, IL1b, IP-10 and RANTES. I am curious about the patients with significant reduction of CECD. Did they have elevated levels of all these cytokines, or some specific ones (different elevation patterns)? Based on Fig 5, some patients had high IL-1b/RANTES and low IL-7/IP-10, and some high IL-7/IP-10 and low IL1b/RANTES. And those without significant CECD loss, they all have low IL1b/IP-10. This is interesting. I believe this paper deserves publication in Plos one.

6. PLOS authors have the option to publish the peer review history of their article (what does this mean?). If published, this will include your full peer review and any attached files.

Reviewer #1: No

---

## [Author Response · Author response to Decision Letter 0]

11 Nov 2021

#Journal Requirements:

Author’s response: Thank you for your comments. We have reviewed the author’s guidelines carefully and confirmed that our manuscript met PLOS ONE's style requirements. 

a) Did participants provide their written or verbal informed consent to participate in this study?

Author’s response: Thank you for your questions. We obtained written informed consent from all participants included in this study.

This study was supported in part by research grants from Japan Science and Technology Agency Center for Revitalization Promotion. 

Author’s response: Thank you for your comments. We removed the funding statement from the manuscript and added it to our revised cover letter.

Reviewer #1: Okabe T et al. evaluated aqueous cytokine levels in the anterior chamber and their correlation with corneal endothelial cells (CEnCs) after DSAEK. This is a well-conducted interesting study and will bring us important information. I think the current manuscript is well-written, however, please consider revising it following the comments below.

Minor comments

1. Title: Please consider revising the title. Title should be simple and specific. The current one is confusing. For example, "corneal endothelial cell transplantation" can be interpreted as "DEMK", "DSAEK", "CEnC cell injection therapy" or "DLEK"... In this case, the authors performed DSAEK, so please specify "Descemet stripping automated endothelial keratoplasty". "anterior aqueous humor" is also confusing. In this article, it is "aqueous humor", as in the text. "reduction" or "loss " would be better than "depletion".

Author’s response to the reviewer’s comments: Thank you for your suggestion. As you advised, we revised the title, as follows:

Association between aqueous humor cytokines and postoperative corneal endothelial cell loss after Descemet stripping automated endothelial keratoplasty

2. Introduction. Well-written with extensive literature review. The hypothesis and purpose are clearly described.

Author’s response to the reviewer’s comments: Thank you for your comments and evaluation. We are very glad to know that you are satisfied with the introduction.

3. Methods: Please clarify whether all patients underwent solitary DSAEK or some of them underwent DSAEK combined with cataract surgery. If so how many?

Author’s response to the reviewer’s comments: Thank you for your suggestion. This is a very important point. We rechecked the data and confirmed that all cases in this study underwent solitary DSAEK. We have added this information to the methods section, as follows:

Line 85:

All cases in this study underwent solitary DSAEK.

4. Tables 1 and 2 can be deleted. The data is well described in the text. The title of table 2 would be "etiologies of bullous keratopathy". "Breakdown" sound strange.

Author’s response to the reviewer’s comments: Thank you for your great suggestion. In response, we changed the contents of Table 1, Table 2 and some parts of the manuscript. We also moved Table 1 and Table 2 from the methods section to the results section and revised the paragraph discussing Table 2 (as below). We also revised the title of Table 2 as you suggested.

Line 181:

Patient data are shown in Table 1. There were significant differences between preoperative visual acuity and visual acuity at 1, 6, and 12 months postoperatively (all P < 0.0001). The etiologies of DSAEK in the studied eyes included pseudophakic bullous keratopathy, post-laser iridectomy bullous keratopathy, FECD, post-trabeculectomy bullous keratopathy, chronic angle-closure glaucoma, corneal endotheliitis, and trauma (Table 2).

5. Methods Line 119. Aqueous sample was frozen without spinning down cellular component, or was it supernatant? Please clarify. Because it sometimes contain immune cells of patients with bullous keratopathy, such as neutrophils, as reported in the article below.

Pathological processes in aqueous humor due to iris atrophy predispose to early corneal graft failure in humans and mice. Sci Adv. 2020 May 13;6(20):eaaz5195.

Author’s response to the reviewer’s comments: Thank you for your question, which is a very important one. In this study, we cryopreserved the samples without centrifugation. We have added this information to the methods sections (in line 125, as below). As you mentioned, it is likely that there were immune cells in the samples, which may have affected our results. We have mentioned this possibility in the discussion and cited the paper you referred to (in line 390, as below).

Line 125:

The samples were frozen without centrifugation at -80° C immediately for further analysis.

Line 398:

It is known that the aqueous humor of eyes with iris atrophy can contain neutrophils and complements.[53 ] In this study, we used aqueous humor samples that did not undergo centrifugation, and this may therefore have affected our results. It is necessary to compare this study with a follow-up study that analyzes the centrifugation-derived supernatant. 

6. Methods: Please add "Data analysis" paragraph, which clarify how the authors defined "iris damage", how they measured "ECD", how they measured VA and so on. "iris damage" is one of the main outcome. In the past reports, iris damage needs careful evaluation. It can be divided into "iris depigmentation", "iris defect" "iris atrophy" and "laser iridotomy". Some previous reports showed mild iris damage (IDS1-2) did not have any influence on CECD after DSAEK, whereas severe iris had serious effect on it. Thus, as in Table 4 and Figures, comparing subjects stratifying them based on the presence of iris damage is not proper, I think. However, the results in the current study is true. So, please add one paragraph in the Discussion section on this matter.

Author’s response to the reviewer’s comments: Thank you for your comments. We appreciate this point very much. We added a “data analysis” paragraph, as suggested. Originally, we included 17 eyes in the iris-damage group and 13 eyes in the non-iris-damage group We rechecked our data and found that the iris-damage group included 16 eyes with damage in one quadrant (laser iridotomy: 11 eyes, peripheral iridectomy: 4 eyes, iris atrophy: 1 eye) and 1 eye with damage in two quadrants (laser iridotomy and peripheral iridectomy). Therefore, it is not likely that iris damage affected the loss of corneal endothelial cells in our patients. We have added this information to the results and discussion sections as follows:

Line 204:

There were 17 eyes with iris damage and 13 eyes without iris damage.

Line 381:

The iris-damage group included 16 eyes with damage in one quadrant (laser iridotomy: 11 eyes, peripheral iridectomy: 4 eyes, iris atrophy: 1 eye) and 1 eye with damage in two quadrants (laser iridotomy and peripheral iridectomy). Therefore, it is not likely that iris damage affected CEC loss.

7. Methods: Please add the information on the donor cornea, domestic or imported, donor age, time between death to preservation, and preservation to surgery if it is available.

Author’s response to the reviewer’s comments: Thank you for your comments. We collected the donor data and summarized them in the results section. The donor corneas were all imported. The mean donor age was 60.5 ± 8.9 years old, the mean time between death and preservation was 762.3 ± 364.4 minutes, and the mean time between preservation and surgery was 6.5 ± 0.5 days. The mean number of donor CECs was 2747.5 ± 217.0 cells/mm2. We checked the correlation between donor age and CEC loss and found that there was no significant correlation. There was also no significant correlation between the preservation period and CEC loss. We added these data to the results section as follows:

Line 187:

The donor corneas were all imported. The mean donor age was 60.5 ± 8.9 years old, the mean time between death and preservation was 762.3 ± 364.4 minutes, and the mean time between preservation and surgery was 6.5 ± 0.5 days. The mean number of donor CECs was 2747.5 ± 217.0 cells/mm2.

Line 193:

There was no significant correlation between CEC loss at 1, 6, and 12 months postoperatively and donor age (P = 0.63, 0.21, 0.42, respectively), time between death and preservation (P = 0.42, 0.51, 0.35, respectively), time between preservation and surgery (P = 0.26, 0.64, 0.59, respectively), or mean number of donor CECs (P = 0.09, 0.10, 0.10, respectively).

8. Results "table 3 title". Please consider rephrasing "inter-period LOSS" not "depletion".

Author’s response to the reviewer’s comments: Thank you for your suggestion. We have revised the title accordingly, as follows:

Table 3. CEC count 1, 6 and 12 months after DSAEK and inter-period loss

9. Discussion is well written. Line 334 "previous reports" need references. The authors found the significant correlations among CECD and cytokines such as IL-7, IL1b, IP-10 and RANTES. I am curious about the patients with significant reduction of CECD. Did they have elevated levels of all these cytokines, or some specific ones (different elevation patterns)? Based on Fig 5, some patients had high IL-1b/RANTES and low IL-7/IP-10, and some high IL-7/IP-10 and low IL1b/RANTES. And those without significant CECD loss, they all have low IL1b/IP-10. This is interesting. I believe this paper deserves publication in Plos one.

Author’s response to the reviewer’s comments: Thank you for your specific advice and interesting comments. We have added the references you requested and have investigated the relationship between IL-7, Il-1b, IP-10 and RANTES in the patients with significant CEC reduction. IL-7 and IP-10 were significantly correlated with each other and were both increased, as shown in the table below. IL-1b and RANTES were also significantly correlated with each other and were also both increased. No significant correlation was observed between these two groups, suggesting that a variety of factors are likely to be involved in CEC reduction.

---

## [Decision Letter · Decision Letter 1]

22 Nov 2021

Association between aqueous humor cytokines and postoperative corneal endothelial cell loss after Descemet stripping automated endothelial keratoplasty

PONE-D-21-21341R1

Dear Dr. Kobayashi,

We’re pleased to inform you that your manuscript has been judged scientifically suitable for publication and will be formally accepted for publication once it meets all outstanding technical requirements.

Kind regards,

Andrew W Taylor, Ph.D.

Academic Editor

PLOS ONE

Additional Editor Comments (optional):

Reviewers' comments:

Reviewer's Responses to Questions

**Comments to the Author**

1. If the authors have adequately addressed your comments raised in a previous round of review and you feel that this manuscript is now acceptable for publication, you may indicate that here to bypass the “Comments to the Author” section, enter your conflict of interest statement in the “Confidential to Editor” section, and submit your "Accept" recommendation.

Reviewer #1: All comments have been addressed

2. Is the manuscript technically sound, and do the data support the conclusions?

Reviewer #1: Yes

3. Has the statistical analysis been performed appropriately and rigorously? 

Reviewer #1: Yes

4. Have the authors made all data underlying the findings in their manuscript fully available?

Reviewer #1: Yes

5. Is the manuscript presented in an intelligible fashion and written in standard English?

Reviewer #1: Yes

6. Review Comments to the Author

Reviewer #1: Thank you for this opportunity to review this article. This is a clinically relevant topic.

Congratulations on your great work.

7. PLOS authors have the option to publish the peer review history of their article (what does this mean?). If published, this will include your full peer review and any attached files.

Reviewer #1: No

---

## [Editor Report · Acceptance letter]

6 Dec 2021

PONE-D-21-21341R1 

Association between aqueous humor cytokines and postoperative corneal endothelial cell loss after Descemet stripping automated endothelial keratoplasty 

Dear Dr. Kobayashi:

I'm pleased to inform you that your manuscript has been deemed suitable for publication in PLOS ONE. Congratulations! Your manuscript is now with our production department. 

Kind regards, 

on behalf of

Dr. Andrew W Taylor 

Academic Editor

PLOS ONE